# Learning to Continually Learn
# with Topological Regularization

**Tananun Songdechakraiwut**                                        SONGDECHAKRA@WISC.EDU
*University of Wisconsin–Madison*

**Xiaoshuang Yin**
*Google*

**Barry D. Van Veen**
*University of Wisconsin–Madison*

**Editors:** Sophia Sanborn, Christian Shewmake, Simone Azeglio, Arianna Di Bernardo, Nina Miolane

## Abstract

Continual learning in neural networks suffers from a phenomenon called catastrophic forgetting, in which a network quickly forgets what was learned in a previous task. The human brain, however, is able to continually learn new tasks and accumulate knowledge throughout life. Neuroscience findings suggest that continual learning success in the human brain is potentially associated with its modular structure and memory consolidation mechanisms. In this paper we propose a novel topological regularization that penalizes cycle structure in a neural network during training using principled theory from persistent homology and optimal transport. The penalty encourages the network to learn modular structure during training. The penalization is based on the closed-form expressions of the Wasserstein distance and barycenter for the topological features of a 1-skeleton representation for the network. Our topological continual learning method combines topological regularization with a tiny episodic memory to mitigate forgetting. We demonstrate that our method is effective in both shallow and deep network architectures for multiple image classification datasets. This extended abstract is adapted from the extended work reported in Songdechakraiwut et al. (2022b)

**Keywords:** Topological data analysis, continual learning

## 1. Introduction

Persistent homology (Edelsbrunner et al., 2000) has emerged as a tool for understanding, characterizing and quantifying the topology of brain networks. Of particular note, Songdechakraiwut et al. (2021, 2022a) employ a 1-skeleton representation for brain networks. The topology of a 1-skeleton is *completely* characterized by connected components and cycles. Here we use persistent homology and the 1-skeleton interpretation of neural networks to improve their performance in continual learning tasks. In particular, we propose a novel *topological regularization* of the neural network's cycle structure to reduce catastrophic forgetting of previously learned task. Regularizing the cycle structure allows the network to explicitly learn its complement, i.e., the modular structure, through gradient optimization. Our approach is made computationally efficient by use of the closed form expressions for the Wasserstein barycenter and the gradient of Wasserstein distance between network cycle structures. We evaluate our approach using image classification across multiple data sets and show that it generally improves classification performance compared to competing approaches in the challenging case of both shallow and deep networks of limited width.

## 2. Efficient Computation of Topology for Network Graphs

**Graph Filtration**   Represent a neural network as an undirected weighted graph $G = (V, \mathbb{W})$ with a set of nodes $V$, and a set of edge weights $\mathbb{W} = \{w_{i,j}\}$. The number of nodes and weights are denoted by $|V|$ and $|\mathbb{W}|$, respectively. Create a binary graph $G_\epsilon$ with the identical node set $V$ by thresholding the edge weights so that an edge between nodes $i$ and $j$ exists if $w_{i,j} > \epsilon$. The binary graph is a simplicial complex consisting of only nodes and edges known as a *1-skeleton* (Munkres, 1996). As $\epsilon$ increases, more and more edges are removed from the network $G$, resulting in a nested sequence of 1-skeletons:

$$G_{\epsilon_0} \supseteq G_{\epsilon_1} \supseteq \cdots \supseteq G_{\epsilon_k},$$

where $\epsilon_0 \le \epsilon_1 \le \cdots \le \epsilon_k$ are called filtration values. This sequence of 1-skeletons is called a *graph filtration* (Lee et al., 2012).

**Birth and Death Decomposition**   The only non-trivial topological features in a 1-skeleton are *connected components* and *cycles*. Persistent homology keeps track of the birth and death of topological features over filtration values $\epsilon$. If a topological feature is born at a filtration value $b_l$ and persists up to a filtration value $d_l$, then this feature is represented as a two-dimensional *persistence* point $(b_l, d_l)$ in a plane. The set of all points $\{(b_l, d_l)\}_l$ is called *persistence barcode* (Ghrist, 2008). The use of the 1-skeleton simplifies the persistence barcodes to one-dimensional descriptors (Songdechakraiwut et al., 2021). Specifically, the representation of the connected components can be simplified to a collection of *birth values* $\mathbb{B}(G) = \{b_l\}$ and that of cycles to a collection of *death values* $\mathbb{D}(G) = \{d_l\}$. In addition, neural networks of the same architecture have a birth set $\mathbb{B}$ and a death set $\mathbb{D}$ of the *same* cardinality as $|V| - 1$ and $|\mathbb{W}| - (|V| - 1)$, respectively. This result completely resolves the problem of point mismatch in persistence barcodes for same-architecture neural networks.

**Closed Form Wasserstein Distance and Gradient**   The Wasserstein distance between 1-skeleton network representations has a closed-form expression. Here we only consider the Wasserstein distance for cycle structure, which depends solely on the death sets. Let $G, H$ be two given networks based on the same architecture. Their (squared) *2-Wasserstein distance for cycles* is defined as the optimal matching cost between $\mathbb{D}(G)$ and $\mathbb{D}(H)$. That is,

$$W_{cycle}^2(G, H) = \min_\phi \sum_{d_l \in \mathbb{D}(G)} \left[ d_l - \phi(d_l) \right]^2,$$

where $\phi$ is a bijection from $\mathbb{D}(G)$ to $\mathbb{D}(H)$. This Wasserstein distance form has a closed-form expression that allows for very efficient computation (Rabin et al., 2011) as

$$W_{cycle}^2(G, H) = \sum_{d_l \in \mathbb{D}(G)} \left[ d_l - \phi^*(d_l) \right]^2,$$

where $\phi^*$ maps the $l$-th smallest death value in $\mathbb{D}(G)$ to the $l$-th smallest death value in $\mathbb{D}(H)$ for all $l$. In addition, the gradient of the Wasserstein distance for cycles $\nabla_G W_{cycle}^2(G, H)$ also has a closed-form expression as (Songdechakraiwut et al., 2022a)

$$\partial W_{cycle}^2(G, H) / \partial d_l = 2 \left[ d_l - \phi^*(d_l) \right].$$

**Closed Form Wasserstein Barycenter** The Wasserstein barycenter is the mean of a collection of networks under the Wasserstein distance and represents the topological centroid. Consider same-architecture networks $G^{(1)}, ..., G^{(N)}$. Let $\mathbb{D}(G^{(i)}) : d_1^{(i)} \leq \cdots \leq d_{|\mathbb{D}|}^{(i)}$ be the death set of network $G^{(i)}$. The *Wasserstein barycenter for cycles* $\mathcal{G}_{cycle}$ has a closed form expression as $\mathcal{G}_{cycle} : \overline{d}_1 \leq \cdots \leq \overline{d}_{|\mathbb{D}|}$, where

$$\overline{d}_l = \sum_{i=1}^{N} \nu_i d_l^{(i)} \Big/ \sum_{i=1}^{N} \nu_i.$$

The complete proof is given in Appendix A.

## 3. Topology Based Continual Learning

Consider a *continual learning* scenario in which $T$ supervised learning tasks are learned sequentially. Each task has a task descriptor $\tau \in \mathbb{T} = \{1, 2, ..., T\}$ with a corresponding dataset $\mathbb{P}_\tau = \{(\boldsymbol{x}_{i,\tau}, \boldsymbol{y}_{i,\tau})_{i=1}^{N_\tau}\}$ containing $N_\tau$ labeled training examples consisting of a feature vector $\boldsymbol{x}_{i,\tau} \in \mathcal{X}$ and a target vector $\boldsymbol{y}_{i,\tau} \in \mathcal{Y}$. We further consider the continuum of training examples that are experienced *only once*, and assume that the continuum is *locally independent and identically distributed* (iid), i.e., $(\boldsymbol{x}_{i,\tau}, \boldsymbol{y}_{i,\tau}) \overset{iid}{\sim} P_\tau$ following the prior work of Lopez-Paz and Ranzato (2017). The goal is to train a model $f : \mathcal{X} \times \mathbb{T} \to \mathcal{Y}$ that predicts a target vector $\boldsymbol{y}$ corresponding to a test pair $(\boldsymbol{x}, \tau)$, where $(\boldsymbol{x}, \boldsymbol{y}) \sim P_\tau$.

Our approach to addressing this problem is to topologically penalize training with future tasks based on the underlying 1-skeleton of the neural network. Define a neural network $G^{(\tau)}$ for learning task $\tau$ with nodes given by neurons, and edge weights defined by the weight/parameter set $\mathbb{W}$. All past-task networks $G^{(j)}$, for $j = 1, ..., \tau - 1$, have the identical node sets with the trained weight set $\mathbb{W}_j^*$ denoting the weights after training through the entire sequence up to task $j$. Since these graphs have the same architecture, their death sets have the same cardinality denoted by $|\mathbb{D}|$. Then the birth-death decomposition of the weight set $\mathbb{W}_j^*$ results in the death set $\mathbb{D}(G^{(j)}) : d_1^{(j)} \leq \cdots \leq d_{|\mathbb{D}|}^{(j)}$. The Wasserstein barycenter for cycles of the first $\tau - 1$ training tasks associated with networks $G^{(1)}, ..., G^{(\tau-1)}$ is

$$\mathcal{G}_{cycle}^{(\tau-1)} : \overline{d}_1^{(\tau-1)} \leq \cdots \leq \overline{d}_{|\mathbb{D}|}^{(\tau-1)},$$

where $\overline{d}_l^{(\tau-1)} = \sum_{j=1}^{\tau-1} \nu_j d_l^{(j)} \Big/ \sum_{j=1}^{\tau-1} \nu_j$. Our approach to learning task $\tau$ minimizes the empirical risk minimization loss (ERM) with the Wasserstein distance and barycenter penalty

$$\mathcal{L}_\tau(\mathbb{W}) = \mathcal{L}_{ERM,\tau}(\mathbb{W}) + \frac{\lambda}{2} W_{cycle}^2(G^{(\tau)}, \mathcal{G}_{cycle}^{(\tau-1)}) \quad \text{for all task } \tau > 1,$$

where $\lambda$ controls relative importance between past- and current-task cycle structure. Intuitively, we penalize changes of cycle structure in a neural network while allowing the network to explicitly learn the modular structure represented by births of connected components.

## 4. Image Classification Experiments

**Datasets** We perform continual learning experiments on four datasets: (1) *Permuted MNIST* (P-MNIST) (Kirkpatrick et al., 2017), (2) *Rotated MNIST* (R-MNIST) (Lopez-Paz and Ranzato, 2017), (3) *Split CIFAR* (Krizhevsky, 2009) and (4) *split miniImageNet* (Russakovsky et al., 2015; Vinyals et al., 2016).

**Network architecture** The P-MNIST and R-MNIST datasets use a fully-connected neural network with two hidden layers each with 128 neurons. The split CIFAR and split miniImageNet datasets use a downsized version of ResNet18 (He et al., 2016) with eight times fewer feature maps across all layers, similar to (Lopez-Paz and Ranzato, 2017).

**Method comparison** We evaluate our method performance in relative to eight baseline approaches that learn a sequence of tasks in a fixed-size network architecture: (1) *finetune*, i.e., a model trained sequentially without any regularization and past-task episodic memory, (2) *elastic weight consolidation* (EWC) (Kirkpatrick et al., 2017), (3) *recursive gradient optimization* (RGO) (Liu and Liu, 2022), (4) *averaged gradient episodic memory* (A-GEM) (Chaudhry et al., 2019a), (5) ORTHOG-SUB (Chaudhry et al., 2020), (6) *experience replay* (Chaudhry et al., 2019b) with *reservoir sampling* (Vitter, 1985) (ER-Res), (7) ER with *ring buffer* (ER-Ring) (Lopez-Paz and Ranzato, 2017) and (8) *multitask* method that jointly learns the entire dataset in one training round and thus is not a continual learning strategy but serves as an upper bound reference for other methods. We employ our proposed topological regularization with reservoir sampling and ring buffer strategies, termed TOP-Res and TOP-Ring.

**Continual learning evaluation** We evaluate the algorithms based on two performance measures: average accuracy (ACC) and backward transfer (BWT) as proposed by Lopez-Paz and Ranzato (2017). Formally, ACC and BWT are defined as

$$\text{ACC} = \frac{1}{T}\sum_{j=1}^{T} R_{T,j}, \quad \text{BWT} = \frac{1}{T-1}\sum_{j=1}^{T-1} R_{T,j} - R_{j,j},$$

where $T$ is the total number of sequential tasks, and $R_{i,j}$ is the accuracy of the model on the $j^{th}$ task after learning the $i^{th}$ task in sequence. We follow the training protocol of Chaudhry et al. (2020).

**Experimental results** Table 1 shows method performance on all four datasets. *Finetune* without any continual learning strategy produces lowest ACC and BWT performance, while the oracle *multitask* is trained across all tasks and sets the upper bound ACC performance for all datasets. Gradient-based RGO and ORTHOG-SUB rely on over-parameterization in a neural network to reduce interference between tasks. Given the long sequence of tasks and small network architecture in our experiments, it is likely that over-parameterization is insufficient for strong performance. Although ORTHOG-SUB achieves highest BWT, it also has the lowest ACC scores, in some cases worse than *Finetune*. ER-Res and ER-Ring achieve high ACC and BWT scores relative to the other baseline methods across all experiments. Our TOP-Res and TOP-Ring methods in turn demonstrate clear performance improvement over ER-Res and ER-Ring, suggesting that our topological continual learning strategy facilitates the consolidation of past-task knowledge beyond that provided by memory replay alone.

Table 1: ACC and BWT performance for different image classification datasets. The mean and standard deviation over five different task sequences are shown.

| | | P-MNIST | | R-MNIST | |
|---|---|---|---|---|---|
| Methods | Memory | ACC (%) | BWT (%) | ACC (%) | BWT (%) |
| Finetune | N | 34.44 ± 2.07 | -58.89 ± 2.23 | 41.43 ± 2.09 | -55.42 ± 2.09 |
| EWC | N | 48.05 ± 1.27 | -44.55 ± 1.46 | 39.80 ± 2.01 | -55.26 ± 2.03 |
| RGO | N | 73.18 ± 0.57 | -19.86 ± 0.61 | 63.34 ± 0.96 | -29.30 ± 0.95 |
| A-GEM | Y | 59.50 ± 1.20 | -33.63 ± 1.24 | 53.38 ± 0.90 | -43.35 ± 0.93 |
| ORTHOG-SUB | Y | 42.83 ± 1.22 | -9.94 ± 1.05 | 23.85 ± 0.84 | -3.45 ± 1.08 |
| ER-Res | Y | 66.38 ± 1.29 | -25.02 ± 1.51 | 72.54 ± 0.36 | -23.48 ± 0.39 |
| ER-Ring | Y | 70.10 ± 0.89 | -23.19 ± 1.01 | 70.52 ± 0.51 | -25.72 ± 0.51 |
| TOP-Res | Y | 68.13 ± 0.66 | -24.50 ± 0.66 | 74.33 ± 0.66 | -20.80 ± 0.63 |
| TOP-Ring | Y | 71.05 ± 0.80 | -22.18 ± 0.86 | 72.12 ± 0.81 | -23.85 ± 0.84 |
| Multitask | – | 90.31 | – | 93.43 | – |
| | | Split CIFAR | | Split miniImageNet | |
| Methods | Memory | ACC (%) | BWT (%) | ACC (%) | BWT (%) |
| Finetune | N | 40.62 ± 5.09 | -23.80 ± 5.31 | 33.13 ± 2.72 | -24.95 ± 2.30 |
| EWC | N | 38.26 ± 3.71 | -25.30 ± 4.57 | 33.48 ± 1.79 | -19.56 ± 2.24 |
| RGO | N | 38.93 ± 1.03 | -18.98 ± 0.89 | 42.03 ± 1.22 | -14.19 ± 1.56 |
| A-GEM | Y | 43.54 ± 6.23 | -23.25 ± 5.65 | 39.52 ± 4.10 | -18.48 ± 4.39 |
| ORTHOG-SUB | Y | 37.93 ± 1.59 | -5.44 ± 1.37 | 32.36 ± 1.44 | -5.52 ± 1.07 |
| ER-Res | Y | 43.28 ± 1.26 | -23.08 ± 1.51 | 38.51 ± 2.40 | -13.58 ± 3.49 |
| ER-Ring | Y | 52.75 ± 1.18 | -14.51 ± 1.79 | 44.67 ± 1.81 | -12.23 ± 1.79 |
| TOP-Res | Y | 45.92 ± 1.50 | -20.10 ± 1.00 | 39.95 ± 1.91 | -14.34 ± 2.06 |
| TOP-Ring | Y | 54.27 ± 1.54 | -11.70 ± 1.27 | 49.08 ± 1.71 | -8.42 ± 1.48 |
| Multitask | – | 61.08 | – | 57.99 | – |

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

## Appendix A. Proof: Closed Form Wasserstein Barycenter

Let $\mathbb{D}(G^{(i)}) : d_1^{(i)} \leq \cdots \leq d_{|\mathbb{D}|}^{(i)}$ be the death set of network $G^{(i)}$. It follows that the $l$-th smallest death value of the barycenter $\mathcal{G}_{cycle}$ of the $N$ networks is given by the weighted mean of all the $l$-th smallest death values of such networks, i.e., $\mathcal{G}_{cycle} : \overline{d}_1 \leq \cdots \leq \overline{d}_{|\mathbb{D}|}$, where

$$\overline{d}_l = \sum_{i=1}^{N} \nu_i d_l^{(i)} \Big/ \sum_{i=1}^{N} \nu_i.$$

**Proof** Recall that the *Wasserstein barycenter for cycles* $\mathcal{G}_{cycle}$ is defined as the death set that minimizes the *weighted* sum of the Wasserstein distances for cycles, i.e.,

$$\mathcal{G}_{cycle} = \arg\min_{\mathcal{G}} \sum_{i=1}^{N} \nu_i W_{cycle}^2(\mathcal{G}, G^{(i)})$$

$$= \arg\min_{\mathcal{G}} \sum_{i=1}^{N} \nu_i \sum_{\overline{d}_l \in \mathcal{G}} \left[ \overline{d}_l - \phi_i^*(\overline{d}_l) \right]^2,$$

where $\nu_i$ is a non-negative weight, and $\phi_i^*$ maps the $l$-th smallest death value in $\mathcal{G}$ to the $l$-th smallest death value in $\mathbb{D}(G^{(i)})$ for all $l$. The sum can be expanded as

$$\sum_{i=1}^{N} \nu_i \sum_{\overline{d}_l \in \mathcal{G}} \left[\overline{d}_l - \phi_i^*(\overline{d}_l)\right]^2 = \sum_{i=1}^{N} \nu_i \left([\overline{d}_1 - d_1^{(i)}]^2 + \cdots + [\overline{d}_{|\mathbb{D}|} - d_{|\mathbb{D}|}^{(i)}]^2\right),$$

which is quadratic. By setting its derivative equal to zero, we find the minimum at $\overline{d}_l = \sum_{i=1}^{N} \nu_i d_l^{(i)} \Big/ \sum_{i=1}^{N} \nu_i.$ ∎

