# OpenReview forum: "Learning to Continually Learn with Topological Regularization"
_NeurIPS.cc/2022/Workshop/NeurReps — NeurReps 2022 Poster_

### Official Review · Reviewer_Mq2M · 2022-10-14
**Review of extended abstract "Topological Continual Learning with Wasserstein Distance and Barycenter"**

**Confidence:** 3
**Soundness:** 3
**Presentation:** 3
**Contribution:** 3
**Overall Rating:** 7

**Summary:**

This article proposes a topological continuous learning method that uses a regularization based on Wasserstein distances from optimal transport, and the addition of episodic memory to prevent forgetting in neural network learning.

**Questions:**

-

**Limitations:**

The submission does not really discuss limitations in enough detail. It could be improved by adding an expanded discussion of this sort.

**Recommended Decision:**

3: Accept

**Relevance:**

4: Highly relevant

**Strengths And Weaknesses:**

The article's idea is interesting, and the presentation is good. I thus recomment acceptance.

The quality of the writing is good in general, but a bit casual in some places. The extended abstract would be improved by redoing the language in this instance:
- Section 2: "Represent a neural network as an undirected ... [...].", "Create [...]."

The mathematics presented at the end of Section 2 and in Section 3 is a bit heavy and sudden. The article could be improved by adding a few more definitions.

**Submission Track:**

Extended Abstract (4 Page)

---

### Official Review · Reviewer_ENzt · 2022-10-14
**Interesting idea with potential, but requires some further work on the analysis and evaluation of results**

**Confidence:** 4
**Soundness:** 3
**Presentation:** 3
**Contribution:** 2
**Overall Rating:** 6

**Summary:**

The authors propose a new topology-based regularization that encourages a neural network to learn modular structure during training by penalizing cycle structure. They have covered the cases for both shallow and deep neural networks of limited width. Finally, they demonstrate this regularization method on Permuted MNIST and Rotated MNIST datasets, using average accuracy (ACC) and backward transfer (BWT) as the perfomance measures.

**Questions:**

- Since the aim only involves detecting graph cycles, could there be potentially simpler graph-theoretic methods that achieve the same aim without invoking topological methods? Why is topological data analysis particularly suitable for this application?
- Why it is desirable for the network to learn modular structure? Perhaps the authors have some justifications in mind, but the paper has not explicitly addressed this in the main text.
- What are the implications of the improvements in terms of ACC and BWT? How to put a few percentage of improvements into context? Are such improvements considered a major progress in the context of the tasks or in the context of continual learning in general?

**Limitations:**

The paper does not discuss limitations, which might be due to page limit.

Here are some suggestions:
- The authors could further develop a more thorough analysis and evaluation of results.
- The results for shallow networks seem to be missing from the paper.
- Some ideas could be better motivated. For example, the authors could elaborate more on:
-- Why is topology particularly suitable for this task? Why not use other graph-theoretic approaches to
-- Why should the network learn modular structure (i.e. why is it good to avoid cycle structures)?
-- How significant are the improvements? What are the implications for continual learning?



**Recommended Decision:**

3: Accept

**Relevance:**

3: Solid fit

**Strengths And Weaknesses:**

Originality:
- (strength) The paper clearly demonstrates originality. It is a very interesting application of topological data anlaysis in continual learning. The connections seem surprising at first but make sense in retrospect. There is a lot of potential in this idea to be developed into a full paper.

Quality:
- (strength) There are both theoretical and numerical results. Some of the details are in the appendix, but overall the results are substantially justified.
- (strength) The numerical results are based on well-established datasets (P-MNIST and R-MNIST).
- (weakness) The paper claims to show that the method applies to both shallow and deep neural networks. However, numerical results are only for neural networks with two hidden layers (deep neural networks).

Clarity:
- (strength) The paper is mostly well-written, with clear statements and explanations for relevant mathematical definitions/theorems.
- (weakness) Certain parts of the paper lack elaborations - particularly the results section. Both the presentation and analysis of results are not sufficiently clear.

Significance.
- (strength) The ideas presented in this paper are very relevant to the theme of the workshop. Including this paper will benefit the discussion at the workshop.
- (weakness) The sections on conclusion and limitations are missing from the main text. (Given the preliminary nature of the workshop, however, this might not be a crucial drawback.)
- (weakness) The paper has never explained how significant the improvements (in terms of ACC and BWT) are the in the context of the tasks or in the context of continual learning in general.

**Submission Track:**

Extended Abstract (4 Page)

---

### Official Review · Reviewer_ov5o · 2022-10-15
**Review of Topological Continual Learning with Wasserstein Distance and Barycenter**

**Confidence:** 3
**Soundness:** 2
**Presentation:** 1
**Contribution:** 3
**Overall Rating:** 4

**Summary:**

The authors propose a method to address the problem of catastrophic forgetting in the context of continuous learning tasks in NN, by regularizing the loss term by a term that enforces the topological structure of the network through its 1-skeleton representation, and is computed using closed forms of the Wasserstein distances and Barycenter.

They test and benchmark their method on various datasets and against state of the art methods, with improved performance reported by adding the regularization to reservoir sampling and ring buffer strategies.

**Questions:**

As already mentioned, the structure of the paper should be much improved for clarity. Appendices should not be written as an augmented version of the text, but section that complement the main text with technical details. They should also be put in order of appearance. Figures should also be put in the main manuscript since they are fundamental to understand and visualize the approach (especially Fig 1 and 2). The paper also crucially lacks of a discussion about the limitations and future works that would help to gain a better perspective on its potential contribution to the field.

I am confused by the results obtained by Orthog-Sub, since  much better performance was obtained on a similar set of tasks as the ones in the paper. (86.6% ACC on permuted MNIST, Chaudhry et al. 2020, as cited in the paper). Can the authors explain the discrepancy?

Why don't the authors also consider the FWT as a metric of performance?

Found few typos or grammar mistakes (but I suspect there could be more): sequantial -> sequential ; we penalize changes (...) while allow -> while allowing p. 13

**Limitations:**

The authors did not address the limitations of their work. More experiments should in particular be conducted to assess the impact of the method, e.g. looking at the performance as the number of tasks increases, convergence of the loss function etc.

**Recommended Decision:**

2: Borderline

**Relevance:**

4: Highly relevant

**Strengths And Weaknesses:**

The authors provide an interesting and original approach by enforcing the topological features of the neural networks in the context of continuous learning, and use a nice mathematical graph representation to do so. While the approach is technically sound, I find the authors perhaps needed more time before the submission to generate more results to support their claims and interpret the current findings. As it probably led the submission to be put in the extended abstract track, it also made the paper tedious to read, with a lot of back and forth between the main sections and appendices, with a structure and arrangement of figures that is suboptimal for clarity. With that said, I believe that with more maturation, this work can lead to a significant contribution to the field.

**Submission Track:**

Extended Abstract (4 Page)

---

### Decision · Program_Chairs · 2022-10-21

Accept (Poster)